# Evolution of Classical and Quantum States in the Groupoid Picture of Quantum Mechanics

**DOI:** 10.3390/e22111292

**Published:** 2020-11-13

**Authors:** Florio M. Ciaglia, Fabio Di Cosmo, Alberto Ibort, Giuseppe Marmo

**Affiliations:** 1Max Planck Institute for Mathematics in the Sciences, 04103 Leipzig, Germany; 2ICMAT, Instituto de Ciencias Matemáticas (CSIC-UAM-UC3M-UCM), Nicolás Cabrera, 13-15, Campus de Cantoblanco, UAM, 28049 Madrid, Spain; fcosmo@math.uc3m.es (F.D.C.); albertoi@math.uc3m.es (A.I.); 3Departemento de Matemáticas, Universidad Carlos III de Madrid, 28911 Leganés, Madrid, Spain; 4Dipartimento di Fisica “E. Pancini”, Università di Napoli Federico II, 80126 Napoli, Italy; marmo@na.infn.it

**Keywords:** quantum mechanics, entanglement, Schwinger’s selective measurements, composite systems, groupoids picture of quantum mechanics, groupoids, Birkhoff–von Neumann logic, foundations of quantum theories

## Abstract

The evolution of states of the composition of classical and quantum systems in the groupoid formalism for physical theories introduced recently is discussed. It is shown that the notion of a classical system, in the sense of Birkhoff and von Neumann, is equivalent, in the case of systems with a countable number of outputs, to a totally disconnected groupoid with Abelian von Neumann algebra. The impossibility of evolving a separable state of a composite system made up of a classical and a quantum one into an entangled state by means of a unitary evolution is proven in accordance with Raggio’s theorem, which is extended to include a new family of separable states corresponding to the composition of a system with a totally disconnected space of outcomes and a quantum one.

## 1. Introduction

Erwin Schrödinger shared with Einstein an enormous puzzlement about the implications of the laws that were being uncovered in the investigation of atomic processes. In their Gedankenexperiment [1], Einstein, Podolski, and Rosen showed the conflicting relation between “Elements of Physical Reality” and the notions of separability and independence in quantum mechanics (see the recent analysis of such a situation in the recent paper [2]). Schrödinger showed his bewilderment in a series of reflections summarized by his famous experiment involving a macroscopic body (a cat) and a quantum system [3], where he argued about the conflict between “common sense” and what we now refer to as an entangled state between a cat and some radioactive material. The experimental construction of entangled states is in general not a trivial matter, and this is why entangled states are considered precious resources in the modern theories known as “resource theories” [4].

In this paper, we will address a problem that underlines some of the previous discussions and that consists of determining whether or not, starting with a composite system made up of a classical and a quantum part, and which is in a separable state, it is possible to build an entangled state by means of a unitary evolution of the system. In the context of the theory of von Neumann algebras, Raggio’s theorem [5] clearly shows that this is not possible, where in that setting, a classical system is described by its algebra of observables, which is an Abelian von Neumann algebra.

We will analyze this question in the groupoid formalism developed recently in [2,6,7,8,9,10]. The result presented here will be that, in accordance with Raggio’s theorem, it is actually impossible to build an entangled state out of a separable state of a classical and a quantum system. This result will crucially depend on the precise meaning of some of the words involved in the previous discussion. Specifically, we need to carefully define what we mean by a “classical” and a “quantum” system, their “composition”, and the “entangled states” of the composite system. In the following sections, we will elaborate on the meaning of these notions in the groupoid formalism, and we will also present an analysis of the logic approach of Birkhoff and von Neumann, in order to show that the standard notion of classical systems coincides with the notion of a classical system in the groupoid formalism, which amounts to the commutativity of the groupoid algebra of the system. Moreover, the formulation of the problem in the groupoid formalism will allow us to present an example that escapes Raggio’s theorem in the sense that, even if both algebras of the systems are non-Abelian, there are separable states that are sent to separable states by all unitary dynamics of the composite system.

## 2. Birkhoff–von Neumann’s Algebra of Propositions

In this section, we will briefly recall the propositional calculus of Birkhoff and von Neumann ([11] and Chapter 5 in [12]). This will serve as a means to investigate the relation between this formalism and the groupoid formalism in the next section and to show that the notion of a classical system coincides in the two formalisms in the sense explained in Theorem 1.

Einstein and Schrödinger were not the only ones having difficulties with the foundational aspects of quantum mechanics. Indeed, also J. von Neumann had his own. His work had an enormous influence because, while providing a solid mathematical background for quantum mechanics (and setting the foundations of the theory of operator algebras), with the aim of clarifying the situation [13], he introduced the idealization of two basic physical operations: measurements and compositions. Without entering here into a discussion on von Neumann’s influence on both notions and our interpretation of them, we would like to point out that, even prior to those, the notion of state worried von Neumann the most. Therefore, we will concentrate on this concept in the present analysis, not because the other two notions are germane to this discussion, but because it will clarify some of the foundational aspects of the groupoid picture of quantum mechanics that were left open in [10].

In order to provide a sound mathematical foundation for quantum mechanics beyond the Hilbert space formalism, J. von Neumann together with G. Birkhoff investigated the logical structure of quantum mechanics. They took an operational point of view that could be summarized in the statement that a system is described by the results of the interrogations we make of it. Thus, they devised a logical system whose propositions P represent experiments performed on the system with just two possible outcomes (yes-no experiments); thus, the system could pass the experiment (unchanged), or it does not pass. There is a natural relation between propositions, denoted P⊆Q, meaning that if the system passes *P*, it will also pass *Q*. As we will see later on, this way of thinking about propositional calculus becomes quite close to Schwinger’s algebra of selective measurements.

After this, a set of axioms, based on their clarity and physical meaning, was proposed characterizing the families of propositions (P,⊆) describing physical systems. Specifically, these axioms are:1the relation ⊆ is a partial order;2P contains two special propositions: *E*, which always allows the passage of the system (tautology), and *∅*, which never permits the passage of the system (absurd);3the pair (P,⊆) is a lattice, i.e., given P,Q∈P, there exists the least upper bound of *P* and *Q*, denoted as P∪Q, and the greatest lower bound, P∩Q;4the lattice (P,⊆) is orthocomplemented, i.e., for any P∈P, there is another proposition $P^{\prime }$ such that:(a)$\emptyset^{\prime }=E$, and $E^{\prime }=\emptyset$;(b)${\left(P^{\prime }\right)}^{\prime }=P$, $P\cap P^{\prime }=\emptyset$, $P\subseteq Q\Leftrightarrow Q^{\prime }\subseteq P^{\prime }$;(c)${\left(P\cup Q\right)}^{\prime }=P^{\prime }\cap Q^{\prime }$.

Finally, it was argued that lattice $\left(P,\subseteq ,\cup ,\cap {,}^{\prime }\right)$ is weakly modular, i.e., if P⊆Q, then *P* and *Q* are compatible, which means that they generate a Boolean sublattice of P (see the discussion in Chapter 5 in [12]). In addition to the previous set of axioms, a Boolean lattice satisfies also the distributive law encoded in the relations:(1)$$\begin{array}{cc} P\cup (Q\cap R) & =(P\cup Q)\cap (P\cup R)\\ P\cap (Q\cup R) & =(P\cap Q)\cup (P\cap R). \end{array}$$
Sometimes, the previous list of the properties of P is supplied with an atomicity condition, which is not relevant for our discussion. In [11], it was proven that the lattice of propositions of a quantum system determines a projective geometry, and propositions can be represented as orthogonal projectors in a Hilbert space, i.e., the standard picture of quantum mechanics in terms of Hilbert spaces is recovered (see also Chapter 8 in [12]).

The propositional calculus discussed so far allows for a clear interpretation of the notion of the state of a quantum system where its statistical interpretation becomes apparent. A state of a quantum system described by a lattice of propositions P is a map $p\;:\;P\;\to \;R_{+}$ satisfying the following axioms:p(∅)=0 and p(E)=1;p(P∪Q)=p(P)+p(Q), if P∩Q=∅.

If the lattice is closed under the joint of countable proposition, these axioms can be extended to their σ-counterpart.

Hence, because of the additivity property of the function *p*, a statistical interpretation of the quantity p(P), as in standard probability theory, as the statistical frequency that the system will pass the experiment *P* in a large series of trials, can be introduced. This constitutes, actually, the formalization of the notion of “Gesamtheit” previously introduced by von Neumann [14] to analyze the statistical and thermodynamical properties of quantum systems. The previous analysis was completed by proving that:(2)$$p\left(P\right)=\mathrm{Tr}\left(\hat{\rho }\hat{P}\right)\;,$$
where $\hat{P}$ is the orthogonal projection associated with *P*, in the Hilbert space H representing the lattice P, and $\hat{\rho }$ is a density operator, i.e., a mixed state, or just a “quantum state” in modern jargon.

## 3. The Groupoid Formalism for Physical Systems

### 3.1. Schwinger’s Selective Measurements and Groupoids

One of the main difficulties with the propositional calculus of Birkhoff and von Neumann was its explicit construction in specific examples. Actually, in spite of his foundational soundness, the propositional calculus developed by Birkhoff and von Neumann was seldomly used to develop the theoretical foundations of the avalanche of experimental data that were obtained at that time (contrary to Dirac’s (often formal) methods, which were so much easier to use).

Almost 20 years later, J. Schwinger produced an axiomatic description of the foundations of quantum mechanics that was somehow related to the propositional calculus of Birkhoff and von Neumann even if no reference was made of the latter. Schwinger’s main idea was to look for a “symbolic language suitable to describe atomic phenomena” [15]. He was able to find the building blocks of such language in the notion of “selective measurement”, which can be considered a sort of enrichment of the notion of proposition or yes-no experiment of the propositional calculus. In particular, denoting by *A* a physical quantity that has the possible values $\left(a,a^{\prime },a^{\prime \prime },\ldots \right)$, the selective measurement M(a) was conceived of as a specific physical apparatus that measures *A* and selects only those physical systems compatible with the value *a* (think, for instance, of a Stern–Gerlach device selecting only the up-beam). This notion was then extended to consider devices capable of accepting systems compatible with the value *a* and changing them into systems compatible with the value $a^{\prime }$. This type of selective measurement is denoted by $M(a^{\prime },a)$, and we clearly see that if *A* has only two possible values, then M(a,a) is a proposition in von Neumann’s sense. However, and this is precisely the main contribution in Schwinger’s conceptualization, instead of considering the family of selective measurements $M(a^{\prime },a)$ as a static kinematical framework, Schwinger added a dynamical interpretation to it by realizing that such selective measurements may be composed in a natural way by performing them one after the other according to the composition rule:(3)$$M\left(a^{\prime \prime },a^{\prime }\right)\;\circ \;M\left(a^{\prime },a\right)\;=\;M\left(a^{\prime \prime },a\right)\;.$$
Then, based on this simple rule, and a few additional natural axioms, Schwinger built an algebra, called the algebra of selective measurements, and to put it to work, he enriched it with a quantum dynamical principle that allowed him to successfully complete his analysis of quantum electrodynamics [16,17,18,19,20,21].

When Schwinger introduced this theoretical framework, the mathematical structure encoded in Equation (Equation 3) was not known to him. However, from a modern perspective, it is immediate to check that the symbolic composition rule given in Equation (Equation 3) satisfies the following axioms:Associativity: $M\left(a^{\prime \prime \prime },a^{\prime \prime }\right)\circ \left(M\left(a^{\prime \prime },a^{\prime }\right)\circ M\left(a^{\prime },a\right)\right)=\left(M\left(a^{\prime \prime \prime },a^{\prime \prime }\right)\circ M\left(a^{\prime \prime },a^{\prime }\right)\right)\circ M\left(a^{\prime },a\right)$;Units: $M\left(a^{\prime },a^{\prime }\right)\circ M\left(a^{\prime },a\right)=M\left(a^{\prime },a\right)$ and $M\left(a^{\prime },a\right)\circ M\left(a,a\right)=M\left(a^{\prime },a\right)$;Inverse: $M\left(a,a^{\prime }\right)\circ M\left(a^{\prime },a\right)=M\left(a,a\right)$ and $M\left(a^{\prime },a\right)\circ M\left(a,a^{\prime }\right)=M\left(a^{\prime },a^{\prime }\right)$.

This implies that the collection of all selective measurements $\left\{M\left(a^{\prime },a\right)\;|\;a,\;a^{\prime }\;\mathrm{outcomes}\;\mathrm{of}A\right\}$ forms a groupoid with the space of objects the measurements $M\left(a,a\right)\equiv 1_{a}$, and Schwinger’s algebra of selective measurements turns out to be the groupoid algebra of this groupoid [22].

Starting from the observation that selective measurements are appropriately described by groupoids, in a series of recent papers [2,6,7,8,9,10], a new picture of quantum mechanics was proposed where Schwinger’s algebra of selective measurements was taken one step forward. In this framework, a physical system is described by means of a groupoid G⇉Ω, where the set Ω is referred to as the space of “outcomes”, and the elements of the groupoid α:x→y, with x,y∈Ω, *x* being the source and *y* the target of α, are referred to as “transitions”. In the following, we will use either the notation G⇉Ω or just G, to denote the groupoid of a physical system.

Conceptually speaking, the family of transitions α∈G generalizes both Schwinger’s notion of selective measurement previously discussed, the actual transitions used in the statement of the Ritz–Rydberg combination principle [23], and the experimental notion of “quantum jumps” introduced in the old quantum mechanics. Furthermore, from a modern perspective, we may also say that the transitions α:x→y represent the abstract notion of amplitudes as “square roots” of probability densities, as argued in [6], that is specific representations of the groupoid G will assign rank-one operators to the transitions α, which will represent the “square roots” of standard probabilities.

### 3.2. The Algebra of Transitions and the Birkhoff–von Neumann Algebra of Propositions

In this section, we will see how to construct an algebra associated with the groupoid G⇉Ω describing a physical system, and we will analyze the relation between this groupoid algebra and the Birkhoff–von Neumann algebra of propositions introduced before. For the sake of simplicity, we will consider discrete, countable groupoids. A similar construction is available for more general groupoids, but requires some additional care to handle functional analytical details [22].

First of all, starting from the groupoid G, we may form the algebra C[G] of formal linear combinations of transitions α:x→y, which is referred to as the algebra of transitions of the groupoid. An element of this algebra will have the form:(4)$$A=\sum_{\alpha \in G}A_{\alpha }\alpha ,\mathrm{with}A_{\alpha }\in C\;,$$
(all $A_{\alpha }$’s zero except for a finite number of them), and the space will be equipped with a natural multiplicative law given by:(5)$$AB=\sum_{\alpha ,\beta \in G}A_{\alpha }B_{\beta }\;\delta_{\left(\alpha ,\beta \right)}\;\alpha \circ \beta \;,$$
where $\delta_{\left(\alpha ,\beta \right)}$ is different from zero only if s(α)=t(β), *s* and *t* being the source and target map of the groupoid, respectively. Clearly, the multiplication above is associative, and in addition, there is a natural involution operation, denoted by * and given by:(6)$$A^{*}=\sum_{\alpha \in G}{\bar{A}}_{\alpha }\alpha^{-1}\;.$$
The involution operation is such that ${\left(AB\right)}^{*}=B^{*}A^{*}$. Elements of this algebra are referred to as “virtual transitions”, and they provide the natural background for a dynamical and statistical interpretation of the theory (see [9,10] for details).

On the other hand, the algebra of transitions also represents a bridge between the groupoid picture of quantum mechanics and the Birkhoff–von Neumann propositional calculus because idempotent elements in C[G] are related to the notion of proposition in the Birkhoff–von Neumann lattice of propositions P associated with a physical system. Indeed, given a groupoid G, an element P∈C[G], such that $P^{*}=P$ and $P^{2}=P$, will be called a proposition of G. Note that, in particular, the units $1_{x}$ with x∈Ω are propositions. There is a natural partial order relation among propositions in G given by:(7)P⊆Q⇔PQ=P.
Note that, if *P* and *Q* are propositions of G and PQ=P, then $PQ=P=P^{*}=Q^{*}P^{*}=QP$. It is immediate to check that the previous relation actually defines a partial order in the space of propositions of G, that is the relation ⊆ thus defined satisfies:Reflexivity: $P^{2}=P$⇒P⊆P;Transitivity: P⊆Q⊆R, i.e., PQ=P and QR=Q, ⇒PR=PQR=PQ=P, i.e., P⊆R;Antisymmetry: P⊆Q and Q⊆P, i.e., PQ=P and QP=Q, ⇒P=PQ=QP=Q, i.e., P=Q.
If the set Ω of outcomes is finite, the algebra of propositions has a unit element $1=\sum_{x\in \Omega }1_{x}$, which may be interpreted as the proposition *E* corresponding to “truth”. Then, we can define an orthocomplement operation $P\;\mapsto \;P^{\prime }$ given by:(8)$$P^{\prime }:=1-P,$$
It is straightforward to check that:$1^{\prime }=\emptyset =0$; $0^{\prime }=1$;P⊆Q⇔$Q^{\prime }\subseteq P^{\prime }$ (as it follows from: $Q^{\prime }P^{\prime }=\left(1-Q\right)\left(1-P\right)=1-P-Q+PQ=1-Q=Q^{\prime }$).
The space of propositions of G would be a lattice if there exists the greatest lower bound (g.l.b.) for any pair (P,Q) of propositions, as is the case if G is finite. However, in the infinite-dimensional case, this is not necessarily so.

In the context of $C^{*}$-algebra theory, it has been recently shown by Marchetti and Rubele [24] that there is a particular type of $C^{*}$-algebras, named Baire*-algebras, or $B^{*}$-algebras, for which the lattice of projection is an orthocomplemented modular lattice. In particular, von Neumann algebras are $B^{*}$-algebras. Therefore, given a groupoid G, if we complete the algebra C[G] of virtual transitions in such a way that it becomes a von Neumann algebra, then its lattice of projections will satisfy the axioms of Birkhoff–von Neumann’s propositional calculus. Notice that, in this setting, the proposition (projection) P∩Q is obtained as:(9)$$P\cap Q\;=\;s-\lim_{n\to \infty }{\left(PQ\right)}^{n}\;,$$
where s-$\lim_{n\to \infty }T_{n}$ denotes the limit of the sequence $T_{n}$ in the strong topology, and it satisfies:(10)$$P\cap P^{\prime }=s-\lim_{n\to \infty }{\left(P\left(1-P\right)\right)}^{n}=0\;.$$
Consequently, setting:(11)$$P\cup Q\;:=\;{\left(P^{\prime }\cap Q^{\prime }\right)}^{\prime },$$
we get an orthocomplemented lattice that will be denoted as P(G). Note that, in addition, this lattice is automatically weakly modular because:(12)P⊆Q⇒PQ=QP=P,
and the sublattice generated by *P* and *Q* is Boolean, that is it is such that:(13)P∪(Q∩P)=(P∪Q)∩PandP∩(Q∪P)=(P∩Q)∪P,
as can be easily checked.

In the particular instance that the groupoid G is countable, as we shall always assume from now on, we may easily complete C[G] to a von Neumann algebra V(G) associated with G as follows. Consider the space $L^{2}\left(G\right)$ of square integrable functions on G with respect to the counting measure. There is an algebra homomorphism λ between the algebra C[G] of transitions of G and the algebra $B(L^{2}\left(G\right))$ given by:(14)$$\left(\lambda \left(A\right)\Psi \right)\left(\beta \right)\;:=\;\sum_{\alpha \in G}\;A_{\alpha }\;\delta_{\left(\alpha^{-1},\beta \right)}\Psi \left(\alpha^{-1}\circ \beta \right)\;,$$
that is $\left(\lambda \left(A\right)\Psi \right)\left(\beta \right)=\;\sum_{\alpha \in G}\;A_{\alpha }\;\Psi \left(\alpha^{-1}\circ \beta \right)$, provided that t(α)=t(β), and zero otherwise. The map λ is called the left-regular representation of C[G] (the notions of the representation of a groupoid and of its algebra are connected with the notions of the representation of a category and of its associated algebra, as explained in [25]). The representation λ is faithful, and thus, in the following, we will often identify C[G] with its image through λ. This choice will simplify the notation.

We define the algebra V(G) as:(15)$$V\left(G\right)\;:=\;{\left(\lambda (C[G])\right)}^{\prime \prime },$$
that is, as the double commutant of λ(C[G]) inside $B(L^{2}\left(G\right))$. Because of von Neumann’s theorem, the algebra $V\left(G\right)\subset B(L^{2}\left(G\right))$ is a weakly closed and strongly-closed subalgebra of the algebra of bounded operators on the separable Hilbert space $L^{2}\left(G\right)$, and thus, it is a von Neumann algebra. The identity operator in V(G) is denoted as 1, and it is not hard to see that $1=\sum_{x\in \Omega }1_{x}$. We refer to V(G) as the von Neumann algebra of the groupoid G.

Thus, we may conclude this section by saying that the set of propositions (projections) of the von Neumann algebra V(G) of virtual transitions of a physical system determined by a countable groupoid G is an orthocomplemented, atomic, weakly-modular lattice of propositions in the sense of Birkhoff–von Neumann’s propositional calculus.

### 3.3. States in the Groupoid Picture

As was discussed in Section 2, from the point of view of the algebra of propositions, a state of a physical system is a non-negative, normalized, σ-additive function *p* on the lattice of propositions P of a physical system. Therefore, in the groupoid picture outlined above, a state *p* will be a normalized non-negative real function on the lattice of propositions P(G) of the von Neumann algebra V(G) of the groupoid G⇉Ω. In other words, ∀P∈P(G), p(1)=1, and p(P)≥0. Moreover, since the lattice of propositions (projections) P(G) generates the total algebra V(G) (essentially because of the spectral theorem), we have that any real element A∈V(G) can be written as:(16)$$A=\sum_{\lambda }\lambda P_{\lambda },$$
where $\left\{P_{\lambda }\right\}$ is the spectral resolution of *A*. Then, $p\left(A\right)=\sum_{\lambda }\lambda p\left(P_{\lambda }\right)$. If *A* is also positive, i.e., $A=B^{*}B$, we will have $p\left(B^{*}B\right)=\sum_{\lambda }{\left|\lambda \right|}^{2}p\left(P_{\lambda }\right)\geq 0$. In other words, states in the sense of von Neumann are just states in the $C^{*}$-algebra V(G), i.e., normalized positive functionals. This justifies the notion of state in the groupoid picture introduced in [10] as normalized, positive functionals defined on the $C^{*}$-algebra of the groupoid G. The case in which *A* has a continuous spectrum may be dealt with in a similar way, essentially “replacing sums with integrals”.

In the case of a countable groupoid, every state ρ determines a function $\varphi_{\rho }\;:\;G\;\to \;C$, which is positive-definite and given by:(17)$$\varphi_{\rho }\left(\alpha \right):=\rho \left(\alpha \right)\;,\;\forall \alpha \in G\;.$$
Note that we have:(18)$$\varphi_{\rho }\left(1_{x}\right)=\rho \left(1_{x}\right)=\rho \left(1_{x}^{*}1_{x}\right)\geq 0,$$
and:(19)$$\sum_{x\in \Omega }\varphi_{\rho }\left(1_{x}\right)=\rho \left(1\right)=1.$$
Hence, the non-negative real numbers,
(20)$$p_{x}=\varphi_{\rho }\left(1_{x}\right)\;\mathrm{with}x\in \Omega \;,$$
define a classical probability distribution on the space of outcomes Ω of the system.

As was shown in [10], any state ρ on V(G), through its associated positive-definite function $\varphi_{\rho }$, defines a decoherence functional on the σ-algebra Σ(G) of parts of G by means of:(21)$$D\left(A,B\right)=\sum_{\begin{array}{c} \alpha \in A,\;\beta \in B\\ t(\alpha )=t(\beta ) \end{array}}\varphi_{\rho }\left(\alpha^{-1}\circ \beta \right)\;,\;A,\;B\in \Sigma \left(G\right)\;.$$
The decoherence functional *D*, in turn, defines a quantum measure, or a Grade-2 measure μ in Sorkin’s conceptualization of the statistical interpretation of quantum mechanics [26], as: (22)μ(A)=D(A,A).
The quantum measure μ nicely captures both the statistical interpretation of experimental observations (it can be interpreted as a statistical frequency), as well as interference phenomena, i.e., it is not additive in general (see [10] and the references therein for a detailed discussion of these subtle aspects). Therefore, in general, we can define the interference function:(23)$$I_{2}\left(A,B\right):=\mu \left(A⊔B\right)-\mu \left(A\right)-\mu \left(B\right)\neq 0\;,$$
where A,B∈Σ(G) are disjoint subsets of G. However, a Grade-2 measure satisfies the following identity:(24)$$I_{3}\left(A,B,C\right):=\mu \left(A⊔B⊔C\right)-\mu \left(A⊔B\right)-\mu \left(A⊔C\right)-\mu \left(B⊔C\right)+\mu \left(A\right)+\mu \left(B\right)+\mu \left(C\right)=0\;,$$
with A,B,C∈Σ(G), three pairwise disjoint subsets of G.

Given two outcomes x,y∈Ω and a state ρ of the system, we can consider the complex number:(25)$$\varphi_{y,x}=\sum_{\begin{array}{c} \alpha :x\to y \end{array}}\varphi_{\rho }\left(\alpha \right)\;,$$
which could be interpreted as the probability amplitude associated with the state ρ and the events *x* and *y* Indeed, we have:(26)$$\begin{array}{cc} |\varphi_{y,x}{|}^{2}={\bar{\varphi }}_{y,x}\varphi_{y,x} & =\sum_{\begin{array}{c} \alpha :x\to y \end{array}}{\bar{\varphi }}_{\rho }\left(\alpha \right)\sum_{\begin{array}{c} \beta :x\to y \end{array}}\varphi_{\rho }\left(\beta \right)=\\ & =\sum_{\begin{array}{c} \alpha :x\to y\\ \beta :x\to y \end{array}}\varphi_{\rho }\left(\alpha^{-1}\right)\varphi_{\rho }\left(\beta \right)\;. \end{array}$$
Then, if the state ρ is factorizable in the sense of [10], i.e., if its associated function $\varphi_{\rho }$ satisfies:(27)$$\varphi_{\rho }\left(\alpha \circ \beta \right)=\varphi_{\rho }\left(\alpha \right)\varphi_{\rho }\left(\beta \right),$$
we obtain that:(28)$$|\varphi_{y,x}{|}^{2}=\sum_{\begin{array}{c} \alpha :x\to y\\ \beta :x\to y \end{array}}\varphi_{\rho }\left(\alpha^{-1}\circ \beta \right)\;,$$
which is the celebrated rule of the “sum-over-histories” composition of the probability amplitudes discovered by R. Feynman.

## 4. Composition of Classical and Quantum Systems

### 4.1. Classical Systems

In Birkhoff–von Neumann’s description of physical systems by means of propositional calculus, classical systems correspond to Boolean lattices. More precisely, two propositions *P* and *Q* are said to be compatible if the sublattice generated by them is Boolean [12], i.e., it satisfies the distributive law in Equation (Equation 13), and a system is classical if all the propositions are compatible among themselves. On the other hand, in the groupoid picture introduced above, we say that a given physical system is classical if the algebra V(G) is Abelian (commutative).

We will now prove that the notion of a classical system from the groupoid point of view corresponds to the notion of a classical system from Birkhoff and von Neumann’s quantum logic point of view and vice versa. First, we will show that if the groupoid algebra V(G) is Abelian, then the lattice of propositions of G is Boolean.

**Proposition** **1.**
*Let G⇉Ω be a countable groupoid with Abelian von Neumann algebra V(G), then its lattice of propositions P(G) is Boolean.*


**Proof.** First, notice that, if V(G) is Abelian, for any pair of propositions *P* and *Q*, we have P∩Q=PQ=QP. Indeed, it clearly holds that ${\left(PQ\right)}^{n}=P^{n}Q^{n}=PQ$, and thus, recalling Equation (Equation 9), it is $P\cap Q=s-\lim_{n}{\left(PQ\right)}^{n}=PQ$. Then, it is:
(29)$${\left(P\cup Q\right)}^{\prime }=P^{\prime }\cap Q^{\prime }=\left(1-P\right)\left(1-Q\right)=1-P-Q+PQ,$$
which means:
(30)P∪Q=P+Q−PQ.
Now, we may easily check the distributive property of the lattice P(G) (see Equation (Equation 1)) by computing:
$$\begin{array}{ccc} P\cup (Q\cap R) & = & P\cap QR=P+QR-PQR\\ & = & P+(PR-PR)+(QP-PQ)+QR-PQR+(-PQR+PQR)\\ & = & (P+Q-PQ)(P+R-PR)=(P\cup Q)(P\cup R)=(P\cup Q)\cap (P\cup R)\;, \end{array}$$
and:
$$\begin{array}{ccc} P\cap (Q\cup R) & = & P(Q\cup R)=P(Q+R-QR)\\ & = & PQ+PR-PQR=(PQ)\cup (PR)=(P\cap Q)\cup (P\cap R)\;. \end{array}$$□

The converse is also true, and in addition, we gain additional information on the structure of the von Neumann algebra of a classical system.

**Theorem** **1.**
*Let G⇉Ω be a discrete countable groupoid. The lattice of propositions P(G) of the groupoid is Boolean if and only if the groupoid G is totally disconnected, its isotropy groups are Abelian, and its von Neumann algebra V(G) is Abelian.*


**Proof.** In Proposition 1, we already saw that if V(G) is Abelian, then P(G) is Boolean. Now, let us prove first that if P(G) is Boolean, then G is totally disconnected. We will do this by showing that, given x≠y∈Ω, there is no transition α between *x* and *y* (i.e., there are no “quantum jumps” from *x* to *y*). Suppose that such a transition α:x→y, with x≠y, exists. Then, let us define the proposition:
(31)$$P_{\alpha }\;:=\;\frac{1}{2}\left(1_{x}+1_{y}+\alpha +\alpha^{-1}\right).$$
Now, notice that we have:
(32)$$1_{x}P_{\alpha }=\frac{1}{2}\left(1_{x}+\alpha^{-1}\right),$$
from which it follows that (see Equation (Equation 9)):
(33)$$1_{x}\cap P_{\alpha }=s-\lim_{\begin{array}{c} n\to \infty \end{array}}{\left(1_{x}P_{\alpha }\right)}^{n}=s-\lim_{\begin{array}{c} n\to \infty \end{array}}\frac{1}{2^{n}}\left(1_{x}+\alpha^{-1}\right)=0\;.$$
On the other hand, we have (see Equation (Equation 11)):
(34)$${\left(1_{x}\cup 1_{y}\right)}^{\prime }=1_{x}^{\prime }\cap 1_{y}^{\prime }=\left(1-1_{x}\right)\cap \left(1-1_{y}\right)=s-\lim_{n}{\left(\left(1-1_{x}\right)\left(1-1_{y}\right)\right)}^{n}=1-\left(1_{x}+1_{y}\right)\;,$$
and hence,
$$1_{x}\cup 1_{y}=1_{x}+1_{y}\;.$$
Moreover,
(35)$$P_{\alpha }\left(1_{x}+1_{y}\right)=2P_{\alpha }.$$
Therefore, we conclude that:
$$P_{\alpha }\cap \left(1_{x}\cup 1_{y}\right)=P_{\alpha }\cap \left(1_{x}+1_{y}\right)=P_{\alpha },$$
and on the other hand, that:
$$\left(P_{\alpha }\cap 1_{x}\right)\cup \left(P_{\alpha }\cap 1_{y}\right)=0\cup 0=0\;,$$
and these two results are obviously different. This means that, if there is a “quantum jump”, we can construct two propositions that are not compatible, and this is a contradiction with the hypothesis of the theorem. Consequently, G⇉Ω is totally disconnected and:
(36)$$G={⊔}_{\begin{array}{c} x\in \Omega \end{array}}G_{x}\;,$$
with $G_{x}$ being the isotropy group at *x*. According to prop. 10.12 in [25], the algebra C[G] of transitions of the totally disconnected groupoid G can be written as:
(37)$$C\left[G\right]\;=\;\underset{x\in \Omega }{⨁}\;C\left[G_{x}\right]\;,$$
where $C[G_{x}]$ is the algebra of transitions of the isotropy group $G_{x}$ seen as a groupoid with only one object. Then, the von Neumann algebra of G will have the form:
(38)$$V\left(G\right)\;=\;\underset{x\in \Omega }{⨁}\;V\left(G_{x}\right)\;.$$
By hypothesis, $G_{x}$ is a countable discrete group, and the lattice of propositions $P(G_{x})$ is a sublattice of the algebra of propositions P(G); hence, it is a Boolean sublattice of the Boolean lattice P(G). The von Neumann algebra $V(G_{x})$ is generated by its projectors, which by hypothesis are compatible propositions; hence, they commute (see Section 5.8 in [12]), and $V(G_{x})$ is Abelian. Hence, all isotropy groups $G_{x}$ must be Abelian, and we have:
(39)$$V\left(G\right)=\underset{\begin{array}{c} x\in \Omega \end{array}}{⨁}V\left(G_{x}\right)≅\underset{\begin{array}{c} x\in \Omega \end{array}}{⨁}L^{\infty }\left({\hat{G}}_{x}\right)=L^{\infty }\left({⊔}_{\begin{array}{c} x\in \Omega \end{array}}{\hat{G}}_{x}\right)\;,$$
with ${\hat{G}}_{x}$ being the Pontryagin’s dual group (or group of characters) of $G_{x}$. □

From this, we conclude that classical systems (at least those described by countable groupoids), both in the groupoid picture and in the quantum logic approach of Birkhoff and von Neumann, correspond to the same notion and are described by Abelian von Neumann algebras that can always be realized as $L^{\infty }\left(X\right)$ with $X={⊔}_{\begin{array}{c} x\in \Omega \end{array}}{\hat{G}}_{x}$. The states of these algebras are non-negative, normalized integrable functions on *X*, i.e., functions $\rho \;:\;X\;\to \;R_{+}$, such that:(40)$$\int_{X}\rho \left(x\right)d\mu \left(x\right)=1\;.$$

### 4.2. Composition

In order to meaningfully discuss the situation depicted in the Introduction, we must consider the composition of two systems, in particular of a classical system with a quantum one. We will carry out the analysis of composite systems in the groupoid framework discussed above.

The composition of systems we will consider here is the simplest one (or the “naive” one), which intuitively corresponds to the idea “to put the systems side by side on the laboratory table” [27,28], i.e., all possible outcomes of both systems can be determined simultaneously, and all possible transitions of both systems can actually happen. As thoroughly discussed in [2], in the language of groupoids, this notion of composition corresponds to the direct product of groupoids. Specifically, given two systems A and B described by two groupoids $G_{A}⇉\Omega_{A}$ and $G_{B}⇉\Omega_{B}$, respectively, their direct composition, denoted by A×B, corresponds to the system described by the groupoid $G_{A}\times G_{B}⇉\Omega_{A}\times \Omega_{B}$, which is the direct product of the groupoids $G_{A}$ and $G_{B}$. The units of $G_{A}\times G_{B}$ are given by the pairs $\left(1_{x_{A}},1_{x_{B}}\right)$, and the inverse of the transition $\left(\alpha_{A},\alpha_{B}\right)$ is ${\left(\alpha_{A},\alpha_{B}\right)}^{-1}=\left(\alpha_{A}^{-1},\alpha_{B}^{-1}\right)$.

Our aim is now to prove that the algebra of transitions of the composite groupoid is the (algebraic) tensor product of the algebras of transitions of the composing groupoids.

**Proposition** **2.**
*Given the countable groupoids $G_{A}⇉\Omega_{A}$, $G_{B}⇉\Omega_{B}$, and $G_{A}\times G_{B}⇉\Omega_{A}\times \Omega_{B}$, it holds that:*
(41)$$C\left[G_{A}\times G_{B}\right]\;≅\;C\left[G_{A}\right]\;⊗\;C\left[G_{B}\right]\;.$$


**Proof.** Consider the algebraic tensor product $C\left[G_{A}\right]\;⊗\;C\left[G_{B}\right]$ of the algebras of transitions of the groupoids, and consider the map $i_{AB}:C\left[G_{A}\times G_{B}\right]\to C\left[G_{A}\right]\;⊗\;C\left[G_{B}\right]$ obtained by extending by linearity the map:
(42)$$\left(\alpha_{A},\beta_{B}\right)\;\mapsto \;i_{AB}\left(\alpha_{A},\beta_{B}\right)\;:=\;\alpha_{A}⊗\beta_{B}.$$
Specifically, denoting by $\left(\gamma_{A},\gamma_{B}\right)$ an element in the direct product $G_{A}\times G_{B}⇉\Omega_{A}\times \Omega_{B}$, every element in the algebra of transition can be written as (see Equation (Equation 4)):
(43)$$a=\sum_{\left(\gamma_{A},\gamma_{B}\right)\in G_{A}\times G_{B}}\;a_{\left(\gamma_{A},\gamma_{B}\right)}\;\left(\gamma_{A},\gamma_{B}\right),$$
and the map $i_{AB}$ reads:
(44)$$i_{AB}\left(a\right)\;=\;\sum_{\left(\gamma_{A},\gamma_{B}\right)\in G_{A}\times G_{B}}\;a_{\left(\gamma_{A},\gamma_{B}\right)}\;\gamma_{A}\;⊗\;\gamma_{B}.$$
It immediately follows that $i_{AB}$ is injective, and we will now see that it is actually surjective. For this purpose, we consider a generic element:
(45)$$c=\sum_{j=1}^{N}\;c_{j}\;a_{A}^{j}\;⊗\;b_{B}^{j}$$
in the algebraic tensor product $C\left[G_{A}\right]\;⊗\;C\left[G_{B}\right]$. Note that *N* is finite and depends on the element *c*. Now, since $a_{A}^{j}\in C\left[G_{A}\right]$ and $b_{B}^{j}\in C\left[G_{B}\right]$, we have:
(46)$$\begin{array}{cc} a_{A}^{j} & \;=\;\sum_{\gamma_{A}\in G_{A}}\;a_{\gamma_{A}}^{j}\;\gamma_{A}\\ b_{B}^{j} & \;=\;\sum_{\gamma_{B}\in G_{B}}\;b_{\gamma_{B}}^{j}\;\gamma_{B}, \end{array}$$
and thus:
(47)$$c=\sum_{j=1}^{N}\;\sum_{\left(\gamma_{A},\gamma_{B}\right)\in G_{A}\times G_{B}}\;c_{j}\;a_{\gamma_{A}}^{j}\;b_{\gamma_{B}}^{j}\;\gamma_{A}\;⊗\;\gamma_{B}\;.$$
Setting:
(48)$$c_{\left(\gamma_{A},\gamma_{B}\right)}\;:=\;\sum_{j=1}^{N}\;c_{j}\;a_{\gamma_{A}}^{j}\;b_{\gamma_{B}}^{j},$$
it is clear that the element:
(49)$$\tilde{c}\;=\;\sum_{\left(\gamma_{A},\gamma_{B}\right)\in G_{A}\times G_{B}}\;c_{\left(\gamma_{A},\gamma_{B}\right)}\;\left(\gamma_{A},\gamma_{B}\right)$$
in $C[G_{A}\times G_{B}]$ is such that
(50)$$i_{AB}\left(\tilde{c}\right)\;=\;c,$$
and thus, $i_{AB}$ is surjective. From this, we conclude that:
(51)$$C\left[G_{A}\times G_{B}\right]\;≅\;C\left[G_{A}\right]\;⊗\;C\left[G_{B}\right]\;,$$
as desired. □

Note that the unit of $C[G_{A}\times G_{B}]$ is given by:(52)$$1_{A\times B}=\sum_{\begin{array}{c} x_{A}\in \Omega_{A}\\ x_{B}\in \Omega_{B} \end{array}}\left(1_{x_{A}},1_{x_{B}}\right)=\left(1_{A},1_{B}\right)\;.$$
If we use the isomorphism $C\left[G_{A}\times G_{B}\right]\;≅\;C\left[G_{A}\right]\;⊗\;C\left[G_{B}\right]$ introduced above, we can write:(53)$$1_{A\times B}=1_{A}⊗1_{B}\;.$$
Note that the left-regular representation of $C[G_{A}\times G_{B}]$ is supported on the Hilbert space:(54)$$L^{2}\left(G_{A}\times G_{B}\right)\;≅\;L^{2}\left(G_{A}\right)\;⊗\;L^{2}\left(G_{A}\right),$$
and thus, it is the tensor product of the left-regular representations of $C[G_{A}]$ and $C[G_{B}]$, respectively.

Now, we will specialize to the composition of a classical system with a quantum system in order to analyze Schrödinger’s Gedankenexperiment in the next section. Hence, if A denotes a classical system with totally disconnected Abelian groupoid:(55)$$G_{A}=\underset{\begin{array}{c} x_{A}\in \Omega_{A} \end{array}}{⨆}G_{x_{A}},$$
where $G_{x_{A}}$ is Abelian $\forall x_{A}\in \Omega_{A}$, and B is a quantum system with groupoid $G_{B}⇉\Omega_{B}$ (the “quantumness” of B is encoded in the assumption that $C[G_{B}]$ is not Abelian), the direct composition of both will be the groupoid $G_{A}\times G_{B}⇉\Omega_{A}\times \Omega_{B}$. The outcomes of the composite system A×B will consist of all possible pairs of outcomes $\left(x_{A},x_{B}\right)\in \Omega_{A}\times \Omega_{B}$, as in the general situation. On the other hand, the transitions of the composition will have the form $\left(\gamma_{x_{A}},\alpha_{B}\right)$ with $\gamma_{x_{A}}\in G_{x_{A}}$ and $\alpha_{B}\;:\;x_{B}\;\to \;y_{B}$. Assuming that $G_{B}$ is connected, the orbits of the groupoid $G_{A}\times G_{B}$ have the form:(56)$$O_{x_{A}}=\left\{\left(x_{A},x_{B}\right)|x_{A}\;\mathrm{fixed},\;x_{B}\in \Omega_{B}\right\}=\left\{x_{A}\right\}\times \Omega_{B},$$
from which we conclude that the groupoid $G_{A}\times G_{B}$ is not connected, and the space of outcomes decomposes as the disjoint union of the family of orbits $O_{x_{A}}$ according to:(57)$$\Omega_{A}\times \Omega_{B}=\underset{\begin{array}{c} x_{A}\in \Omega_{A} \end{array}}{⨆}O_{x_{A}}.$$

As before, the algebra of virtual transitions of the composite system A×B has the form:(58)$$C\left[G_{A}\times G_{B}\right]\;=C\left[G_{A}\right]\;⊗\;C\left[G_{B}\right]\;,$$
while, to understand the structure of the von Neumann algebra $V(G_{A}\times G_{B})$, we need a preliminary lemma.

**Lemma** **1.**
*Let $H_{A}$ and $H_{B}$ be two complex, separable Hilbert spaces, and let A and B be two unital *-subalgebras of $B(H_{A})$ and $B(H_{B})$, respectively. Let $V\left(A\right)\subset B\left(H_{A}\right)$ and $V\left(B\right)\subset B\left(H_{B}\right)$ denote the von Neumann algebras generated by A and B, respectively (i.e., their double commutants or weak closures). Then, it holds that:*
(59)$$V\left(A\right)\hat{⊗}V\left(B\right)=V\left(A⊗B\right),$$
*where $\hat{⊗}$ denotes the tensor product of von-Neumann algebras.*


**Proof.** We will consider first the algebraic tensor product A⊗B of *A* and *B*. The resulting algebra is supported on the Hilbert space $H=H_{A}⊗H_{B}$. We denote by $V\left(A⊗B\right)={\left(A⊗B\right)}^{\prime \prime }$ the von Neumann algebra generated by it. Clearly, V(A⊗B) is supported on H. The von Neumann algebra $V\left(A\right)\hat{⊗}V\left(B\right)$ is itself also supported on H, essentially by the definition of the tensor product of von Neumann algebras [29].Now, consider the canonical inclusion $A⊗B\subset V\left(A\right)\hat{⊗}V\left(B\right)$. The closure in the weak topology will induce a continuous inclusion:
$$V\left(A⊗B\right)\subset V\left(A\right)\hat{⊗}V\left(B\right)\;.$$
On the other hand, there are natural inclusions: A↪A⊗B, $a↪a⊗1_{B}$ and B↪A⊗B, $b↪1_{A}⊗b$, that induce inclusions V(A)⊂V(A⊗B), V(B)⊂V(A⊗B), respectively. Consequently, V(A)⊗V(B)⊂V(A⊗B). Finally, considering the closure with respect to the weak operator topology, we get:
$$V\left(A\right)\hat{⊗}V\left(B\right)\subset V\left(A⊗B\right)\;.$$□

Now, consider the separable Hilbert spaces $H_{A}=L^{2}\left(G_{A}\right)$, $H_{B}=L^{2}\left(G_{B}\right)$, and $H=H_{A}⊗H_{B}$ and the unital *-subalgebras $C[G_{A}]$ and $C[G_{B}]$ of $B(H_{A})$ and $B(H_{B})$, respectively. Recalling Equation (Equation 58) and applying Lemma 1, we obtain:(60)$$V\left(G_{A}\;\times \;G_{B}\right)\;=\;V\left(G_{A}\right)\;\hat{⊗}\;V\left(G_{B}\right)\;.$$
Moreover, the algebra $C[G_{A}]$ of the classical system reads (recall the discussion in Section 4.1 and prop. 10.12 in [25]):(61)$$C\left[G_{A}\right]\;=\;\underset{x_{A}\in \Omega_{A}}{⨁}\;C\left[G_{x_{A}}\right],$$
and thus:(62)$$V\left(G_{A}\right)\;=\;\underset{x_{A}\in \Omega_{A}}{⨁}\;V\left(G_{x_{A}}\right)\;.$$
Exploiting the distributive property of the tensor product over the direct sum, we obtain:(63)$$V\left(G_{A}\times G_{B}\right)\;=\;\underset{x_{A}\in \Omega_{A}}{⨁}\;\left(V\left(G_{x_{A}}\right)\;\hat{⊗}\;V\left(G_{B}\right)\right)\;.$$

It is important to point out that the direct product structure of the algebra of the composite system is manifest only because of the groupoid formalism used in its construction. Indeed, in the algebraic approach, one starts with an Abelian algebra for the classical system and with a non-Abelian algebra for the quantum system and then builds the algebra of the composite system as the tensor product of the algebra. The result is an algebra in a form similar to that in Equation (Equation 58) for the algebra of the direct product of groupoids. However, it is clear that the direct sum decomposition of the tensor product algebra given in Equation (Equation 63) explicitly depends on the fact that the groupoid of the classical system is disconnected, and thus, if we have access only to the information on the algebra of the classical system (i.e., the fact that it is an Abelian algebra), we are not able to immediately detect and appreciate the direct sum decomposition. This instance points to the fact that the groupoid formalism allows a more comprehensive understanding of the properties of physical systems and their associated algebras in a way that is conceptually similar to how the algebraic picture of quantum mechanics provides a more comprehensive understanding of the Hilbert space picture (think, for instance, of the notion of superselection sectors in Chapter IV.1 in [30]).

## 5. Separable States and Unitary Evolution

We are ready now to consider the problem of the evolution of separable states. Specifically, we consider a composite system A×B, where *A* is a classical system and *B* is a quantum system, and a closed evolution of the composite system that reflects the fact that the system does not interact with the external world. We need to understand first what happens to the composite system when the initial state is a product state.

From the point of view of the theory of von Neumann algebras, it is known that the space of states of a von Neumann algebra, which is the tensor product of an Abelian algebra with an arbitrary one, does not contain entangled states because of Raggio’s theorem [5] (see also [29,31,32]). Therefore, if the system A is considered to be a classical system in the sense discussed in the previous sections, then, given an arbitrary initial product state, it is not possible to create an entangled state by means of a dynamical evolution.

In the rest of this section, we will offer an alternative view of Raggio’s theorem based on the groupoid picture introduced before. The main point will be the use of the direct product decomposition of the algebra given in Equation (Equation 63) and of the properties of the representations of the groupoid $G_{A}\times G_{B}$ to show that there is no unitary dynamics of the composite system that can produce an entangled state from a product one. Of course, this result is expected in the light of Raggio’s theorem; however, the derivation we offer is completely independent of Raggio’s theorem. Moreover, the strategy we adopt to prove our result will allow presenting an example in which a conclusion similar to that of Raggio’s theorem is true even if both algebras are non-Abelian. Specifically, if one of the two groupoids, say $G_{A}$, is totally disconnected, but its isotropy subgroups are no longer Abelian, then the resulting algebra will be non-Abelian, but there will be a particular family of product states on $V(G_{A}\times G_{B})$ that cannot evolve into entangled states by means of a unitary dynamics.

Given a composite system A×B, where *A* is a classical system (i.e., it is described by a totally disconnected groupoid $G_{A}$ with Abelian isotropy subgroups and Abelian von Neumann algebra $V(G_{A})$), we know from the discussion in the previous section that the groupoid $G_{A\times B}$ of the composite system is the direct product $G_{A}\times G_{B}$ of the groupoids $G_{A}$ and $G_{B}$, and the von Neumann algebra $V(G_{A}\times G_{B})$ has the form written in Equation (Equation 63).

A closed dynamical evolution is given by a strongly continuous one-parameter group $\Phi_{t}$ of automorphism of $V(G_{A}\times G_{B})$ [9]. For the purpose of this paper, we consider dynamical evolutions $\Phi_{t}$ that can be written in terms of a one-parameter group $u_{t}=\exp itH$ of unitary elements in $V(G_{A}\times G_{B})$ by means of:(64)$$\Phi_{t}\left(a\right)\;=\;u_{t}\;a\;u_{t}^{†}\;.$$
This is true if $G_{B}$ is connected, in which case, it gives rise to a non-Abelian von Neumann algebra $V(G_{B})$. Equation (Equation 64) is the Heisenberg picture of the dynamics, while the Schrödinger picture can be obtained by taking the dual action of group $\Phi_{t}$ on the dual of $V(G_{A}\times G_{B})$. Specifically, if ρ is a state on $V(G_{A}\times G_{B})$, then the evolution $\rho_{t}$ of the state will be given by:(65)$$\rho_{t}\left(a\right)\;=\;\rho \left(\Phi_{t}\left(a\right)\right)\;=\;\rho \left(u_{t}\;a\;u_{t}^{†}\right)$$
for all $a\in V(G_{A}\times G_{B})$.

We will build a family of vector states by using the so-called fundamental representation $\pi_{0}$ of the groupoid $G_{A}\times G_{B}$ of the system and of its algebra $V(G_{A}\times G_{B})$. Here, we limit ourselves to recalling only those aspects of the fundamental representation that are strictly needed in the following and refer to [8,25] for further details.

The fundamental representation is supported on the separable Hilbert space $H_{A\times B}$ given by:(66)$$H_{A\times B}=L^{2}\left(\Omega_{A}\times \Omega_{B}\right)≅L^{2}\left(\Omega_{A}\right)⊗L^{2}\left(\Omega_{B}\right)\equiv H_{A}⊗H_{B}\;.$$
Note that the vectors $|x_{A},x_{B}\rangle =|x_{A}\rangle ⊗|x_{B}\rangle$ corresponding to the delta function at $\left(x_{A},x_{B}\right)$ on $\Omega_{A}\times \Omega_{B}$ determine an orthonormal basis of $H_{A\times B}$. The representation of a transition $\alpha :\left(y_{A},y_{B}\right)⟶\left(z_{A},z_{B}\right)$ in $G_{A}\times G_{B}$ on a basis element $|x_{A},x_{B}\rangle$ is given by:(67)$$\pi_{0}\left(\alpha \right)|x_{A},x_{B}\rangle \;=\;\delta_{x_{A}y_{A}}\;\delta_{x_{B}y_{B}}|z_{A},z_{B}\rangle .$$
Given an element $a=\sum_{j}a^{j}\alpha_{j}$ in $V(G_{A}\times G_{B})$ (where the sum may be infinite provided it converges in the weak topology), we have:(68)$$\pi_{0}\left(a\right)|x_{A},x_{B}\rangle \;=\;\sum_{j}\;a^{j}\pi_{0}\left(\alpha_{j}\right)|x_{A},x_{B}\rangle \;.$$

On the other hand, the Hilbert space $H_{A\times B}$ decomposes as the direct integral:(69)$$H_{A\times B}=\int_{\Omega_{A}}^{⊕}H_{x_{A}}\;d\mu \left(x_{A}\right)\;,$$
where every:(70)$$H_{x_{A}}=C|x_{A}\rangle ⊗H_{B}\;$$
is an invariant subspace, that is we have $\pi_{0}\left(a\right)H_{x_{A}}\subset H_{x_{A}}$ for every $a\in V(G_{A}\times G_{B})$ (note also that $H_{x_{A}}\;≅\;H_{B}$). Actually, for any finite subset $\Delta \subset \Omega_{A}$, we get the (finite-dimensional) subspace $H_{\Delta }\subset H_{A\times B}$ given by:(71)$$H_{\Delta }=\underset{\begin{array}{c} x_{A}\in \Delta \end{array}}{⨁}H_{x_{A}}\;.$$
Since $G_{A}$ is totally disconnected (*A* is a classical system), Equation (Equation 68) leads to:(72)$$\pi_{0}\left(a\right)|x_{A},x_{B}\rangle =\sum_{j}a^{j}|x_{A},y_{Bj}\rangle \;,$$
where $a=\sum_{j}a^{j}\alpha_{j}$ with $\alpha_{j}=\gamma_{Aj}⊗\alpha_{Bj}$, $\gamma_{Aj}\in G_{x_{A}}$, and $\alpha_{Bj}:x_{B}\to y_{Bj}$, and to:(73)$$\pi_{0}\left(a\right)|x_{A},x_{B}\rangle =0$$
if $\gamma_{Aj}\in G_{x_{Aj}}$ with $x_{Aj}\neq x_{A}$ for all *j*.

This means that the fundamental representation $\pi_{0}$ has a block diagonal structure graphically depicted as: (74)π0=⋱πxAπxA′πxA″⋱,
where $\pi_{x_{A}}=\pi_{0}{|}_{H_{x_{A}}}$ or, in other words: $\pi_{0}={⨁}_{x_{A}\in \Omega_{A}}\pi_{x_{A}}$.

Consider now the normalized vector $|\Psi_{x_{a}}\rangle \in H_{A\times B}$ given by:(75)$$|\Psi_{x_{a}}\rangle \;=\;|x_{A},\psi_{B}\rangle \;=\;|x_{A}\rangle \;⊗\;|\psi_{B}\rangle \;.$$
with $|\psi_{B}\rangle \in H_{B}$. This vector defines a state $\rho_{x_{A}}$ on $V(G_{A}\times G_{B})$ by means of:(76)$$\rho_{x_{A}}\left(a\right)\;:=\;\langle \Psi_{x_{a}}|\pi_{0}\left(a\right)|\Psi_{x_{a}}\rangle ,$$
and $\rho_{x_{A}}$ is clearly a product state on $V(G_{A}\times G_{B})$, of a classical-quantum composite system. We will now analyze its dynamical evolution $\Phi_{t}^{♯}\left(\rho_{x_{A}}\right)$ and show that it is a product state for every *t*.

Indeed, the Heisenberg evolution represented in $H_{A\times B}$ takes the form:(77)$$\pi_{0}\left(\Phi_{t}\left(a\right)\right)\;=\;\pi_{0}\left(u_{t}\;a\;u_{t}^{†}\right)\;=\;\pi_{0}\left(u_{t}\right)\;\pi_{0}\left(a\right)\;\pi_{0}\left(u_{t}^{†}\right),$$
and we immediately conclude that:(78)$$\left(\Phi_{t}^{♯}\left(\rho_{x_{A}}\right)\right)\left(a\right)\;=\;\langle \Psi_{x_{A}}|\pi_{0}\left(u_{t}\right)\;\pi_{0}\left(a\right)\;\pi_{0}\left(u_{t}^{†}\right)|\Psi_{x_{A}}\rangle \;.$$
Then, recalling that the subspace given in Equation (Equation 70) is invariant with respect to $\pi_{0}$, we have that:(79)$$\pi_{0}\left(u_{t}^{†}\right)|\psi \rangle \;=\;\pi_{0}\left(u_{t}^{†}\right)\left(|x_{A}\rangle ⊗|\psi_{B}\rangle \right)\;=\;|x_{A}\rangle ⊗|\psi_{B}^{t}\rangle ,$$
which means that $\rho_{x_{A}}=\Phi_{t}^{♯}\left(\rho_{x_{A}}\right)$ is a product state for all *t*, as claimed. Furthermore, we may consider separable states of the form:(80)$$\varrho \;=\;\sum_{x_{A}\in \Omega_{A}}\;p_{x_{A}}\;\rho_{x_{A}}$$
with $\rho_{x_{A}}$ as in Equation (Equation 76) and with $p_{x_{A}}\geq 0$ such that $\sum_{x_{A}\in \Omega_{A}}p_{x_{A}}=1$, and we immediately obtain that $\varrho_{t}=\Phi_{t}^{♯}\left(\varrho \right)$ is a separable state for all *t*. Note that the same statement holds if we start with a mixed state both for the quantum system or the classical system.

The results of this section are summarized in the following theorem:

**Theorem** **2.**
*Let A be a system described by a totally disconnected, countable groupoid $G_{A}⇉\Omega_{A}$. Let B be a system described by a countable, connected groupoid $G_{B}⇉\Omega_{B}$. Let A×B be the composite system described by the direct product groupoid $G_{A}\times G_{B}⇉\Omega_{A}\times \Omega_{B}$. Given a dynamical evolution of A×B described by a one-parameter group $\Phi_{t}$ of automorphisms of the von Neumann algebra $V(G_{A}\times G_{B})$ of the system that can be written according to Equation (Equation 64), assume that at the initial time $t_{0}$, the system is in the separable state ϱ given by Equation (Equation 80). Then, the dynamical evolution of ϱ will be a separable state for all t.*


It is important to note that this result depends crucially on the form of the groupoid algebra of a composite system made of a classical and a (possibly) quantum system, and of its fundamental representation. Indeed, we see that what is actually needed in the argument outlined above is the fact that the subspace given in Equation (Equation 70) is invariant with respect to the fundamental representation $\pi_{0}$. This instance is realized even if the groupoid $G_{A}$ is a countable groupoid that is totally disconnected, but presents non-Abelian isotropy subgroups. In this case, the algebra $V(G_{A})$ is no longer Abelian (the system *A* is thus non-classical), but the argument given above still applies.

## 6. Conclusions

In this contribution, the composition of classical and quantum systems in the groupoid formalism for quantum systems is analyzed. As a first step, the notion of classical systems is discussed, and for countable groupoids, it is found that a classical system in the sense of the quantum logic of Birkhoff and von Neumann is necessarily associated with a totally disconnected groupoid with Abelian isotropy subgroups, and vice versa. Then, it is asked what happens to the evolution of a separable state of a composite system made up of a classical system and a quantum one. The arguments presented in this paper show that, under the appropriate technical conditions expressed in Theorem 2, the direct composition of a classical system with a quantum one prevents a separable state of the composite systems from evolving into an entangled state by means of unitary evolution. As said before, if the system *A* is classical in the sense specified in Section 4.1, the content of Theorem 2 is a particular case of Raggio’s theorem. However, when the isotropy subgroups of *A* are non-Abelian, then the algebra $V(G_{A})$ is non-Abelian, and Raggio’s theorem does not apply, while Theorem 2 does apply and allows us to conclude that there are separable states of the composite system that will remain separable under all possible unitary dynamics of the composite system. Therefore, even if both systems are non-classical, there still are particular initial separable states of the composite system for which Schrödinger’s disturbing evolution is precluded. These type of states may be called “pseudo-classical” states, and Theorem 2 represents a first step stating clearly that pseudo-classical states actually exist. In the context of quantum information theory, a similar result has already been obtained [33]. In this case, both the systems A and B are described by connected groupoids (the groupoids of a pair of n elements giving rise to matrix algebras), and it is proven that there are mixed separable states that remain separable under every unitary dynamics. The set of such states (which are pseudo-classical in the sense specified above) is referred to as the Gurvits–Barnum ball, and it would be interesting to understand the proof given in [33] in the context of the groupoid picture. All these issues will be dealt with in future works.

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
