# Peer review of "Evolution of Classical and Quantum States in the Groupoid Picture of Quantum Mechanics"

_entropy, 2020, doi:10.3390/e22111292_

Round 1

Reviewer 1 Report

The authors still phrase their results in term of a cat in a pure state. That is pure nonsense. A cat has 10^26 atoms and even more degrees of freedom. Schrodinger's original idea has been corrected by going to density matrices. All such endless repetitions of the cat story are misconceptions about what QM stands for.

Now the authors do derive several results, which I do not doubt. To save the case, they should stress that the it is standard parlor to do it this way, though it is utter nonsense by itself. This should be done in intro and recalled in the conclusion.

Author Response

We re-submit a revised version of our old paper “Schrödinger’s problem with cats: measurements
and states in the Groupoid Picture of Quantum Mechanics” (entropy-873301), where the comments and recommendations of both the editor and referees have been taken into account. The title and introduction have been rewritten stating precisely the scope and intent of the paper. Thus all references to the Schrödinger’s cat experiment has been removed and we leave the readers to extract their own conclusions from the results presented in the paper.

Reviewer 2 Report

The authors explain why my objection to the first submission was based on a misunderstanding and they clear up this misunderstanding. This is fine as far as I'm concerned. The manuscript should be checked by a physicist, not a philosopher of physics.

Author Response

We re-submit a revised version of our old paper “Schrödinger’s problem with cats: measurements
and states in the Groupoid Picture of Quantum Mechanics” (entropy-873301), where the comments and recommendations of both the editor and referees have been taken into account. The title and introduction have been rewritten stating precisely the scope and intent of the paper. Thus all references to the Schrödinger’s cat experiment has been removed and we leave the readers to extract their own conclusions from the results presented in the paper.

This manuscript is a resubmission of an earlier submission. The following is a list of the peer review reports and author responses from that submission.

Round 1

Reviewer 1 Report

The authors claim that they resolve the quantum measurement problem, aka the Schroedinger cat paradox, in the groupoid picture of quantum mechanics. However, this claim is unfounded, since it is not admissible to presuppose classical states from the outset. The issue is how to account for measurement outcomes by basing oneself on quantum states. The authors are not familiar with the state of the art of the discussion about the measurement problem in the foundations of physics. A good starting point is the paper by Tim Maudlin, Three measurement problems, in Topoi 1995, which is generally acknowledged today as offering a precise formulation of the problem and setting the standard for its solution.

Reviewer 2 Report

The paper describes a novel approach, based on the groupoid formalism of quantum mechanics, to Schrödinger's paradox. The material is not quite self-contained, but it is nevertheless explained in a satisfactorily pedagogical way. The end result is a resolution of the paradox, showing that products between classical states and quantum states cannot evolve into entangled states under unitary and isolated time evolution. This is a striking conclusion, following a relatively straightforward construction (albeit one that relies on very specific technical assumptions), so it seems to definitely deserve publication.

Reviewer 3 Report

This paper misses the point and should not be published.

The problem with the Schrodinger cat is the wish to describe a macroscopic system (the cat) as a pure state. This is completely unrealistic.

To get a pure state, one has to purify a mixed state by doing ideal measurements and selecting the subensemble related to a specific pointer reading. This can be achieved in practice for a handful of variables. A cat has some 10^26 degrees of freedom. Nobody can ever do 10^26 measurements and selections. Hence pure states for macroscopic objects don't exist.

Probably the whole idea of the paper rests on the broadly spread, false idea that "every system has a wave function".

Quantum mechanics, however, deals with ensembles for which Born's rule gives probabilities for measurement outcomes. The discussion of Schrodinger cats proves that "the wavefunction of the system" is a nonsense. It does not exist and it is needed for nothing else but confusion.

So the description of a macroscopic cat by a pure state is a mistake, the formulas you write down have nothing to do with the system you are thinking of.
